# The First Evidence on the Occurrence of Bisphenol Analogues in the Aqueous Humor of Patients Undergoing Cataract Surgery

**DOI:** 10.3390/jcm11216402

**Published:** 2022-10-29

**Authors:** Jolanta Flieger, Tomasz Śniegocki, Joanna Dolar-Szczasny, Wojciech Załuska, Robert Rejdak

**Affiliations:** 1Department of Analytical Chemistry, Medical University of Lublin, Chodźki 4A, 20-093 Lublin, Poland; 2Department of Pharmacology and Toxicology, National Veterinary Research Institute, 24-100 Pulawy, Poland; 3Department of General and Pediatric Ophthalmology, Medical University of Lublin, Chmielna 1, 20-079 Lublin, Poland; 4Department of Nephrology, Medical University of Lublin, Jaczewskiego 8, 20-090 Lublin, Poland

**Keywords:** bisphenols, bisphenol A, bisphenol F, bisphenol A isomers, aqueous humor, ultrahigh-pressure liquid chromatography (UHPLC)

## Abstract

Human exposure to BPs is inevitable mostly due to contaminated food. In this preliminary study, for the first time, the presence of bisphenols (BPs) in aqueous humor (AH) collected from 44 patients undergoing cataract surgery was investigated. The measurements were performed using a sensitive ultra-high performance liquid chromatography-tandem mass spectrometry (UHPLC–MS/MS). Chromatographic separation was achieved using a reverse-phase column and a gradient elution mode. Multiple reaction monitoring (MRM) was used. The method was validated for bisphenol A (BPA) and bisphenol F (BPF). The limits of quantification (LOQs) of both investigated analytes were 0.25 ng mL^−1^. The method was linear in the range of 0.25–20.0 ng mL^−1^ with correlation coefficients (R^2^) higher than 0.98. Recovery of analytes was in the range of 99.9 to 104.3% and intra-assay and inter-assay precision expressed by relative standard deviations (RSD%) were less than 5%. BPA was detected in 12 AH samples with mean concentrations of 1.41 ng mL^−1^. BPF was not detected at all. Furthermore, two structural isomers termed BPA-1, and BPA-2 were identified, for the first time, in 40.9% of the AH samples, with almost twice higher mean concentrations of 2.15 ng mL^−1^, and 2.25 ng mL^−1^, respectively. The total content of BPs were higher in patients with coexisting ocular pathologies such as glaucoma, age-related macular degeneration (AMD), and diabetes in comparison to cataracts alone. However, the difference between these groups did not reach statistical significance (*p* > 0.05). Performed investigations indicate the need for further research on a larger population with the aim of knowing the consequences of BPs’ accumulation in AH for visual function.

## 1. Introduction

The aqueous humor (AH) of the eye is the fluid located in the anterior chamber of the eye. Despite the fact that the volume of AH is very small and in humans it amounts to about 0.3–0.5 mL, it is necessary for the proper functioning of the vision organ. AH is formed by diffusion and ultrafiltration of plasma; however, the composition of the fluid is modified due to the secretion of the unpigmented ciliated epithelium [1,2,3]. The composition of AH includes immunoglobulins, neuropeptides, proteins, lipids, carbohydrates, enzymes, vitamins, and microelements. Previous studies are usually devoted to the search for the relationship between the composition of AH and various eye diseases such as age-related macular degeneration (AMD), cataracts, diabetic retinopathy (DR), glaucoma, etc. [4,5,6]. Most of these works concern the analysis of cataract progression. For this purpose, the composition of amino acid and lipid metabolites such as glutaric acid and pelargonic acid [7], markers of lipid peroxidation [8], enzymatic antioxidants such as superoxide dismutase (SOD) and catalase (CAT) [8], and tryptophan metabolites [9] were investigated. Due to the small amount of material for research, restrictions related to research on humans, and trace amounts of analytes in AH, advanced analytical techniques are used for analysis, such as nano-LC-ESI-MS/MS [10], ultra-high-performance liquid chromatography coupled with triple quadrupole mass spectrometry with electrospray ionization (UHPLC-ESI-MS/MS) [9], ICP-OES [11], ICP-MS [12].

AH analysis has not yet been fully utilized in the diagnosis of eye diseases. There is a lot to be investigated, for instance in the area of xenobiotic risk or environmental toxin exposure assessment in the context of searching for causes of visual impairment. So far, there are no studies on the search for bisphenols (BPs) in AH. It is known that there are many sources of exposure to these environmental toxins as they are widely used in the production of polycarbonate plastics and epoxy resins [13]. The most common of them is bisphenol A (BPA; 4,4′-isopropylidenediphenol). BPA migrates to the environment as evidenced by reports of its detection in various environmental matrices such as soil, water, air, and plants [14,15]. Human exposure to BPA occurs primarily through the oral route, via food and drink packed in plastic containers. However, there is the possibility of exposure to BPs by transdermal or inhalation routes. Due to high lipophilicity (log *P* = 3.4), BPA can easily cross biological barriers such as blood–placental and blood–brain [16]. As a result of constant exposure, humans may accumulate BPA in tissues from which it is gradually released [17]. Previous studies have shown that BPA can be accumulated in the brain, causing toxic effects [18] that manifest as nervous and behavioral disorders, impairment of visual memory (object recognition) and spatial memory (object localization) [19]. Unfortunately, the neural function of BPA, particularly at the level of the visual neurons, remains poorly understood. The presence of BPA has been detected in human blood. However, the obtained results differ from each other depending on the patient’s living place. The mean serum level in Japan was in the range of 0.33 to 0.59 ng/mL (*n* = 20) [20], whereas in Hong Kong it was 0.95 ng/mL (*n* = 153) [21].

To measure BPs in biological matrices, different analytical techniques are applied. Some of them are immunochemical techniques such as, e.g., enzyme-linked immunosorbent assay (ELISA) used for the determination of BPA in serum samples [22,23]. However, this test is characterized by low sensitivity and specificity due to the high probability of interaction with interfering substances present in the matrix [22]. The commercially available kit for the determination of the content of BPA in biological samples has a measurement range of 0.3–100 ng/mL [24]. Gas chromatography coupled with mass spectrometry (GC-MS) is frequently used for the determination of BPA in various types of matrices. Due to its low volatility, BPA requires derivatization with acetic anhydride, N-O-bis (trimethylsilyl) trifluoroacetamide (BSTFA), pentafluorobenzyl bromide (PFBBr), or 4-(4,5-diphenyl-1-imidazoyl chloride) benzoyl (DIB-Cl) [25]. High-performance liquid chromatography (HPLC) is also used for the determination of BPA. HPLC can be combined with various detectors such as: spectrophotometric (UV-Vis, ultraviolet-visible detector) [26], fluorescent (FLD, fluorescence detector) [27], electrochemical detector (ED) [28], mass spectrometer [29], or the tandem mass spectrometer (MS/MS) [30]. BPA was determined in various biological matrices, e.g., serum [31,32], urine [33,34,35,36,37,38,39], food [28,40,41,42], amniotic fluid [43,44,45], breast milk [46].

Many studies have demonstrated the toxic effects of BPs on human health. It should be remembered that BPA is an endocrine-disrupting compound (EDC). That is why human exposure to BPA affects reproduction [47], thyroid function [48], hypertension, diabetes, and obesity [49,50]. According to the United States Environmental Protection Agency (US EPA), the acceptable BPA dose is 50 µg/kg body weight/day [51]. The World Health Organization (WHO) estimated the average consumption of BPA present in food products to be <0.04–0.40 µg/kg/day, and in the worst case this value could be 1.40 µg/kg [52,53]. Since BPA is detected in the urine of most people, there is no doubt that exposure to this compound is common today.

In literature, there are no studies confirming the involvement of BPA in human eye pathology, although it is known that BPA intoxication is associated with high blood pressure (HBP) [54], which is one of the key factors in retinopathy. Akintunde, et al. [55] used BPA as an agent of ocular impairment in a rat model to explore ocular protein aberration and its associated cataract. Another study is the work of Xu et al. [56] describing the disturbance of the visual function of cats, through the disruption of individual activity of neurons and the processing of visual information in the primary vision path as a result of exposure to BPA. The authors of the study suggest that BPA-induced degeneration of visual perception is the result of changes in the subcortical regions. Evidence of this claim is the increased expression of phosphorylated regulated extracellular kinase (ERK) observed in BPA-exposed neurons [57]. ERK is considered to be a factor that influences the modulation of synaptic vesicle mobilization and the triggering of the excitatory transmitter [58].

In our preliminary study, an attempt was made to quantify BPA and bisphenol F (BPF) in the AH samples collected during cataract surgery. BPF is a BPA analog also belonging to endocrine-disrupting compounds. BPF is widely used as an alternative to BPA in the production of various products such as food packaging, sealants, and dental composites. The analysis was performed by UHPLC/MS/MS and included 44 patients. To the best of our knowledge, this is the first attempt to determine BPs in this biological material.

## 2. Materials and Methods

### 2.1. Standards and Solvents

The bisphenols’ standards: 2,2-Bis(4-hydroxyphenyl)propane-d_16_ (Bisphenol A-d_16_, (IS)), 4,4′-methylenediphenol (bisphenol F—BPF), 2,2-Bis(4-hydroxyphenyl)propane, 4,4′-Isopropylidenediphenol (bisphenol A, BPA) were obtained from Sigma-Aldrich (Bellefonte, PA, USA). Methanol (MeOH), isopropanol (*iso*-PrOH), and formic acid (HCOOH) of LC-MS grade were purchased from E. Merck (Darmstadt, Germany). LC-MS grade water was purchased from Sigma-Aldrich (St. Louis, MO, USA). All analytical solvents and types of equipment were checked for BPs’ contamination prior to experiments. Individual stock standard solutions were prepared in methanol and stored in screw-capped glass tubes in a refrigerator (+4 °C) under the same conditions for up to two weeks.

### 2.2. Sample Collection

The AH samples were taken from the patients undergoing surgical procedures, namely, cataract phacoemulsification in the Chair and Department of General and Pediatric Ophthalmology, Medical University of Lublin. The study was approved by the Local Bioethical Committee of the Medical University of Lublin (approval no. KE-0254/154/06/2022). Aqueous fluid extraction was performed at the microscope through paracentesis at the first stage of the cataract removal procedure. A 27- or 30-gauge needle was used, with the extraction of 0.1 to 0.2 mL of AH followed by injection of a balanced salt solution. After collection, the samples were immediately frozen and stored in 1.5-mL polypropylene tubes at −80 °C until analysis. The samples were collected in bisphenol-free tubes. The study group consisted of 26 women, and 18 men (Table 1). Some patients, in addition to cataracts, suffered from coexisting diseases: seven people suffered from diabetes, ten from age related macular degeneration (AMD), and five from glaucoma.

### 2.3. Sample Preparation

A constant amount of the internal standard (IS) was added to all samples. To 45 µL of the sample (some samples were rejected because there was not enough material) 5 µL of (IS) was added. Then, 10 μL was directly injected into LC-MS/MS.

### 2.4. LC-MS/MS

The UHPLC-MS/MS system consisting of an AB Sciex ExionLC UHPLC system connected to an AB Sciex API 5500 Qtrap mass spectrometer (AB Sciex, Concord, ON, Canada). Analyst 1.6.3 software (AB Sciex) controlled the UHPLC-MS/MS system and Multiquant 3.2 (AB Sciex) was used to process the data. The mass spectrometer was operated in the negative ESI mode with a capillary voltage of −4.5 kV. The temperature of desolvation was set at 500 °C, curtain gas (N_2_)–35; collision gas (N_2_)–high; gas 1 (air)–40; gas 2 (air)–40. The multiplier was set at 2300 V. The flow rate of mobile phase was 400 μL min^−1^, the injection volume was 10 μL. The chromatography was performed on a Synergi 4 μ Fusion-RP column (50 mm × 2 mm × 4 μm), connected to a C18 precolumn (4 mm × 2 mm × 4 μm). The mobile phase for LC analysis consisted of two solutions: A (0.5% isopropanol in 0.1% formic acid in water) and B (methanol). The mobile phase gradient program started at 20% of B, 80% B at 1.0 min to 4.7 min, then 20% of B at 5.0 min and held for 2.0 min as equilibration step. The column operated at 40 °C and the ions were monitored in Multiple Reaction Monitoring (MRM) mode (Table 2).

### 2.5. Validation

The analytical-method validation was carried out according to the criteria accepted for bioanalytical method validation. The ICH Q2 (R1) method-validation protocols [59,60] and other requirements [61,62,63] were taken into account. The following validation parameters were determined: selectivity, matrix effect, linearity, limits of detection (LOD) and quantification (LOQ), precision, accuracy, and recovery.

#### 2.5.1. Calibration Standards and Quality Control Samples

Stock standard solutions (1 mg mL^−1^) and working solutions (0.5; 2.0; 5.0; 20.0; 40.0 μg mL^−1^) BPA, and BPF were prepared in methanol and stored at −20 °C until analysis. The internal standard was BPA-d_16_ at a concentration of 1 mg mL^−1^ (standard stock solution) in methanol, and working solutions were prepared daily at the concentration of 5 μg mL^−1^. Calibration standards ranging from the limit of quantification (LOQ) to 20.0 ng mL^−1^ of AH were individually prepared daily to check linearity by adding suitable amounts of methanol working solutions to pre-checked bisphenol A free AH fluids. Quality Control (QC) samples at three concentrations 0.25; 2.5; 20.0 ng mL^−1^ (low, medium, high) were also prepared daily to check validation parameters e.g., accuracy, precision, recovery, etc.

#### 2.5.2. Linearity

The quantification was performed by the internal standard method. The same amount of the internal standard was included in each of the calibration standards ensuring its final concentration of 2.5 ng mL^−1^. Spiked blank samples were prepared as follows: 5 μL of the standard solution of BPA and BPF at different concentrations were added to 40 μL of the bisphenol-free AH and 5 μL of internal standard (IS) at a concentration of 5 µg mL^−1^ was added. The prepared mixtures were subjected to the UHPLC procedure. The internal standard calibration curves were obtained by plotting the ratio of the analyte peak area to the internal standard peak area against the analyte concentration. Linearity was determined by the least-squares regression method. Acceptable linearity was achieved when the coefficient of determination was at least 0.98. The resultant calibration curve was applied to calculate the amount of analyte present in the investigated sample.

#### 2.5.3. Precision and Accuracy

Precision and accuracy were determined at five concentration levels 0.25; 1.0; 2.5; 10.0; 20.0 ng mL^−1^ by analyzing six different daily replicates of samples giving intraday precision values and six replicates along three subsequent working days giving intermediate precision values, both expressed as the coefficient of variation (RSD%) expected to be less than 20%.

#### 2.5.4. LOD, LOQ

LOD and LOQ were estimated by measuring replicates (*n* = 20) of a blank sample. The standard deviation (SD) of the mean noise level over the retention time window of each analyte was determined. The LOD values were calculated as the mean value +3 (SD), whereas LOQ as the mean +6 SD.

#### 2.5.5. Matrix Effects, Recovery

Matrix effects, recovery, and process efficiency were determined using the experimental design proposed by Matuszewski et al. [64]. Set 1 was five replicates of QC solutions prepared in the mobile phase. Set 2 was five different replicates of blank samples fortified with QC solutions. Matrix effects were determined by dividing the mean peak areas of set 2 by set 1 multiplied by 100. Recovery was determined by comparing the mean peak areas of compounds under investigation obtained in set 2 to those in set 1 multiplied by 100.

### 2.6. Statistical Analysis

In order to determine descriptive statistics for data sets in this study Excel Data Analysis ToolPak 2016 was applied. Statistical analyses were performed in the R statistical environment (R Core Team 2016) [65], as well as R language libraries [66,67,68,69]. The pronounced right-handed asymmetries of the BPs’ concentration distributions in the analyzed groups were observed. Due to the fact that the assumptions considering normality and homogeneity of variance were not met in all sets of data, the Wilcoxon–Mann–Whitney test was used for intergroup comparisons [70,71].

## 3. Results

### 3.1. UHPLC–MS/MS Analysis

Wilczewska et al. [72] described troubles of the determination of BPA at ultra-trace levels by liquid chromatography and tandem mass spectrometry. The authors suggested that the low repeatability of the BPA analysis is caused by the mobile phase contamination with BPA. Authors proved that BPA contaminating the mobile phase is able to enrich in the front of the column when mobile phase elution strength is too low. The problem of BPA originating from the mobile phase was solved by preferring the isocratic elution mode with the mobile phase of relatively high elution strength. The authors recommended using 50% of acetonitrile as the mobile phase. In our study, we propose methanol because it allows better separation than acetonitrile. Additionally, based on our previous experience, a small amount of isopropanol was added because we have observed that the isopropyl alcohol possessing a higher elution strength in RP systems reduces the peak tailing of peaks making them slightly higher [73].

The applied procedure was based on a “dilute-and-shoot” LC-MS/MS methodology used in toxicology [74,75]. This method protects against the potential degradation of BPs or contamination as a result of the complex extraction of the sample. UHPLC ensured and improved the throughput and sensitivity of the method. The elaborated conditions ensured successful separation of examined analytes within 5 min (Figure 1, Figure 2 and Figure 3).

Fragmentation of BPA in the mass spectrometer showed two main product ions (fragments m/z 133, and 212) at m/z 227.2 as the precursor ion. Therefore, these MRM transitions were chosen to identification BPA in AH samples. These product ions have been previously used to quantify BPA by MS analysis [76,77,78] as ions with relatively higher abundance.

### 3.2. Validation Protocol

A procedure based on UHPLC-MS/MS has been developed and validated for the quantification of BPA and PBF in AH. The isotope labeled BPA (bisphenol–D16) internal standard added at the stage of sample preparation, ensured improvement of the precision and accuracy of the quantitative determination due to eliminating the matrix effect, effects related to loss of analyte, or different sample volumes [79].

Representative chromatograms obtained by spiking BPA-free, and BPF-free AH by the standards at a concentration of 0.25 ng mL^−1^ are shown in Figure 1a,b and Figure 2a,b, respectively. Figure 3 shows the chromatogram of the internal standard.

Quantification of target analytes were investigated by an internal standard method at five concentration points. Linear calibration curves showed determination coefficients (R^2^) equal to or higher than 0.98 in all cases. The limit of detection (LOD) and the limit of quantification (LOQ) values were adequate for the purpose of the present study (Table 3). The intra- and inter-assay precision and recovery were in accordance with the internationally established acceptance criteria. The calculated RSD% for repeatability and reproducibility showed values that were always lower than 5% (Table 4).

### 3.3. Concentrations of BPs in AH Samples

The AH samples taken from 44 Polish participants undergoing cataract surgery were collected and analyzed. Three peaks were resolved in the MRM transition m/z 227.2/212. This indicates that instead of BPA, two of its structural isomers existed in the AH samples, and were labeled BPA-1, and BPA-2 in increasing order of retention time. The compound eluted at a retention time of 4.58 min was BPA, which was confirmed using the BPA standard. BPA isomers were identified when at least two selected MRM transitions responded at the same retention time. Two additional peaks observed, exhibited the same fragment pathways as BPA, and they both showed m/z 227.2/133, and 227.2/212 fragments. Representative UHPLC–MS/MS chromatograms are shown in the Figure 4.

The raw data for the male/female cataract subjects with/without coexisting ophthalmic pathology is presented in Appendix A. Around 40% of collected whole AH samples contained detectable BPA-1 (red) and BPA-2 (green) (Table 5). More precisely BPA was detected in 12 AH samples (27.27%), with mean concentrations of 1.39 ng/g. BPF was not detected at all. The structural isomers of BPA (BPA-1, BPA-2) were identified in 18 samples (40.91%), for the first time in AH, at mean concentrations of 2.15 ng/mL and 2.25 ng/mL, respectively. The concentration profiles of ∑BPs (sum of BPA-1, BPA-2, BPA) in AH samples is shown in Figure 5. The graph shows only these subjects exhibited any presence of BPs.

### 3.4. Comparison between the Specific Groups

All clinical samples have been divided into different groups: one group included patients with cataracts and the second one with cataracts and coexisting ocular diseases like AMD, glaucoma, and the third one with coexisting diabetes. Furthermore, we compared males and females in order to determine whether there is statistical evidence that the associated populations are significantly different according to total BPs content in AH samples.

In the group of women (F), a clear right-sided asymmetry was observed, therefore the Mann–Whitney test was used to compare selected groups by gender. Descriptive statistics for the male and female groups are presented in Table 6, and Figure 6. A higher median value of BPs concentration (2.21 ng mL^−1^) was found in the group of men compared to the group of women (0.41 ng mL^−1^). However, obtained correlation did not achieve a statistical significance level of *p* = 0.09 (*p* > 0.05).

The statistical analysis relating to the comparison of the cataract group with the group of patients with additional ocular pathologies, i.e., cataract, glaucoma, and AMD, is presented in Table 7 and Figure 7. Also, in this case, it can be noticed that while the median value is much higher for the group with cataract and coexisting eye diseases (1.90 ng/mL) compared to the cataract group (0.5 ng/mL), the difference between these groups did not reach statistical significance (*p* = 0.18).

Including diabetes as an additional disease showed differences in median values but did not change the statistical significance of the compared groups (*p* > 0.05) (Table 8, Figure 8).

To compare two independent data sets coming from distinct populations, which do not affect each other, the Mann–Whitney test being the nonparametric counterpart of the *t*-Student test, has been applied. Obtained values of *p* indicate that there were differences in the median values regarding the level of total BPs in AH samples. but they were not statistically significant. The total content of BPs was higher in patients with coexisting ocular diseases (glaucoma and AMD) and diabetes in comparison to cataracts alone. It should be emphasized that nonparametric tests such as the Mann–Whitney test have poor discriminatory power. Therefore, we cannot exclude the possibility that the results could turn out significant if replicated with a larger cohort group.

## 4. Discussion

Our study is the first to measure BPs in human AH samples and the first to show the presence of free BPA, as well as two of its structural isomers, assigned as BPA-1 and BPA-2. Although it is already known that BPA exists in the form of several structural isomers, identified as by-products in industrial production, only a few studies described the occurrence of BPA isomers in samples of human origin. The first report on the occurrence of BPA isomers tentatively termed B_1-4_-BPA was published in 2022 by Li et al. [80]. The authors also analyzed three BPA metabolites such as BPA-sulfate, BPA-glucuronide, and BPA- bisulfate disodium salt. The method used in this study, however, required a multi-step sample preparation procedure to enable the analysis of human serum and whole blood by UHPLC-MS/MS.

In our study, not even one form of BPA was detected in about 1/3 of the respondents. It is very difficult to indicate the reasons why BPs are detected in some individuals while others do not have them at all. To explain this, it would be necessary to know what the exposure to BPs was or conduct more extensive examinations of the health of patients with the aim of pointing out any metabolic differences between them. In our opinion, this stage of research should be transferred to animal experiments, where it is possible to systematically monitor the BPs’ exposure or the activity of key enzymes. It is worth noticing that the levels of BPA isomers were approximately two times higher in comparison to BPA. Patients undergoing cataract surgery usually suffered from additional diseases such as diabetes, hypertension, glaucoma, and AMD. The BPs detected in AH seem to be only one of the environmental factors contributing to the deterioration of their health. It should be emphasized, however, that patients who, in addition to cataracts, suffered from other ocular diseases such as AMD, and glaucoma, or diabetes had higher mean total levels of BPs. The men’s group was also characterized by elevated total levels of BPs concentration in AH samples in comparison to the women’s group.

Demonstrating the presence of BPA in AH seems to be important from the point of view of many potential threats related to BPs intoxication starting from direct oxidative damage [81], changing the activity of many enzymes [82,83,84,85,86,87,88,89,90,91], deregulation of the gene expression [55,92], up to the increase in pro-inflammatory cytokines [93], and apoptotic proteins [94].

BPA can work by changing the activity of many enzymes, including the increase in the activity of the angiotensin-converting enzyme (ACE), i.e., the HBP marker [82,83]. It is known that arterial hypertension causes abnormal blood flow and, consequently, visual disturbances and retinal damage, related to the development of hypertensive retinopathy [84], which predisposes to the development of cataracts. BPA works not only through the renin–angiotensin–aldosterone system (RAAS), but also by blocking the activity of ocular endothelial nitric oxide synthase (eNOS) [85]. NO is a key molecule that has the ability to dilate blood vessels and regulate microcirculation. Decreased NOS expression has been associated with the development of hypertensive retinopathy, cataracts, glaucoma, vitreous hemorrhage [86], and the cell proliferation (hyperplasia) of ocular cancer and degeneration of retinocyte outgrowth, with the deterioration of conjunctiva and lens [87]. Another enzyme that is associated with BPA intoxication is arginase, the increased expression of which can trigger inflammatory processes [88] that lead to the development of cataracts. Another enzyme whose activity increases under the influence of BPA is phosphodiesterase-5 (PDE-5). PDE-5 breaks down cyclic guanylate monophosphate (cGMP) necessary for the relaxation of optical vessels. Thus, an increase in PDE-5 activity causes endothelial dysfunction of the eye. The increase in the activity of acetylcholinesterase (AChE), butyrylcholinesterase (BuChE) and monoamine oxidase-A (MAO-A) is also responsible for the occurrence of HBP and an increased risk of developing ocular hypertension and related complications. It has been observed that the decreased activity of NOS after exposure to BPA also increases the hydrolysis of ATP by the enzymes ATPase, ADPase, AMPase and the activity of adenosine deaminase (ADA) in the eye cells in arterial hypertension in rats [89]. It is particularly dangerous for the development of oculopathy and the predisposition to ischemic injury to the optic nerves [90]. It should be emphasized that intracellular adenosine has been described as an endothelial-derived hyperpolarization factor due to the ability to relax and hyperpolarize vessels smoothing the muscle cells [91]. A study by Akintunde, et al. [55] shows that exposure to BPA increases the expression of the CD43 gene in the eye, which regulates T-cell function. This fact indicates that BPA interferes with the immune function of the eyes [92]. BPA also increases the expression of pro-inflammatory cytokines (tumor necrosis factor TNF-α., interleukins, IL-1ß) which are involved in oncogenesis and cause cell inflammation [93], damage to the cornea, lacrimal gland, retinopathy, and dry keratoconjunctivitis (KCS). BPA intoxication increases the level of apoptotic proteins, including caspase-9, in the eye of rats, which suggests an increased risk of developing eye cancer following eye trauma [94]. There is also a danger that an increase in caspase-9 expression with low NO levels in retinocytes, activating cGMP hydrolysis, may induce degeneration upon exposure to ultraviolet B radiation [95]. Also, the transcription of the gene encoding P53 (tumor protein) increases after exposure to BPA [96].

Moreover, it should be remembered that BPA is primarily a xenoestrogen. Considering the fact that estrogen receptors (ERs) are also located in the nodes of the visual pathway, such as the retina [97,98] and the primary visual cortex [99], BPA, even in low doses can affect the functioning of the eye by antagonizing estrogenic functions [100].

Simultaneous exposure to BPA and its isomers complicates the prediction of potential hazards. To our knowledge, there is still no data concerning isomers’ toxicity therefore we are not able to predict their influence on the vision organ. Data on isomers should therefore be required in the assessment of exposure to BPs, especially as they occur at almost twice the concentration level.

## 5. Conclusions

The BPs (BPA, and BPF) are the most commonly applied chemical plasticizers used in the production of plastics to enhance their rigidity. Our preliminary study reports the presence of BPA and its isomers in AH of patients undergoing cataract surgery. The validated UHPLC–MS/MS method was applied for the quantification of the analytes. BPA was detected in only some samples, whereas the presence of BPF at a level higher than the detection limit was not confirmed. Besides BPA, its two structural isomers tentatively named BPA-1, and BPA-2 at almost two times higher concentrations in comparison to BPA, were found. The considerable amount of BPs from a few to several ng per mL of AH may represent the degree of exposure of human eyes to these environmental toxins.

Confirmed accumulation of BPs in AH may be the real threat to the proper function of the visual organs, especially since the total concentration of BPs in AH of patients suffering from cataracts and coexisting diseases was higher in comparison to cataract patients. Our research shows that men are more likely to accumulate BPs. In view of the potential risks outlined in the discussion, there is a dire need to continue the research in a larger patient population to shed new light on BPA-induced ocular dysfunctions.

## Figures and Tables

**Figure 1 jcm-11-06402-f001:**
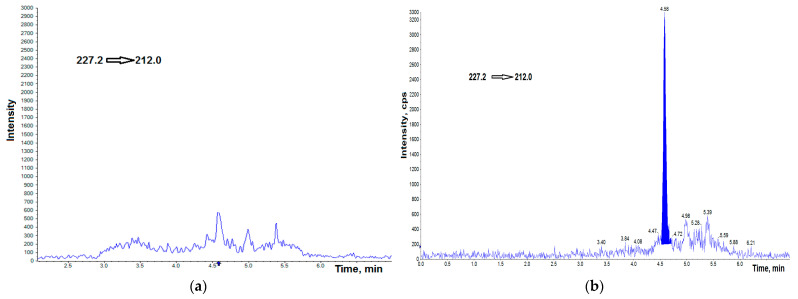
UHPLC–MS/MS chromatogram of BPA-free HA (**a**), BPA-free HA spiked with 0.25 ng g^−1^ BPA (**b**). cps—counts per second.

**Figure 2 jcm-11-06402-f002:**
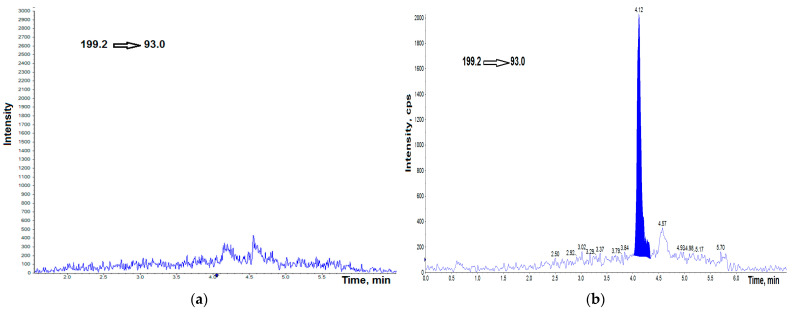
UHPLC–MS/MS chromatogram of BPF-free HA (**a**), BPF-free HA spiked with 0.25 ng g^−1^ BPF (**b**). cps—counts per second.

**Figure 3 jcm-11-06402-f003:**
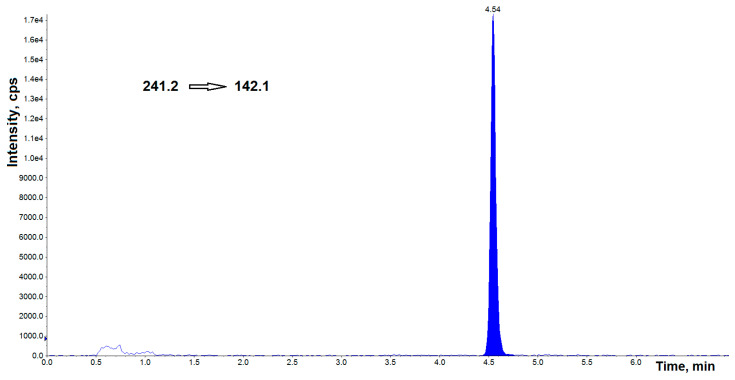
UHPLC–MS/MS chromatogram of bisphenol A-d_16_ (IS) at concentration of 2.5 ng mL^−1^. cps—counts per second.

**Figure 4 jcm-11-06402-f004:**
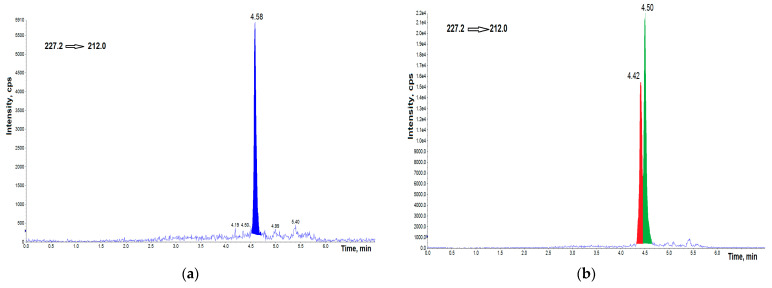
UHPLC–MS/MS chromatogram of AH samples with detected BPA (**a**), AH sample with two detected isomers of BPA (**b**), AH sample with detected BPA and two isomers (**c**), AH sample with BPA and two detected isomers and spiked with 2.0 ng g^−1^ of BPA (**d**). cps—counts per second.

**Figure 5 jcm-11-06402-f005:**
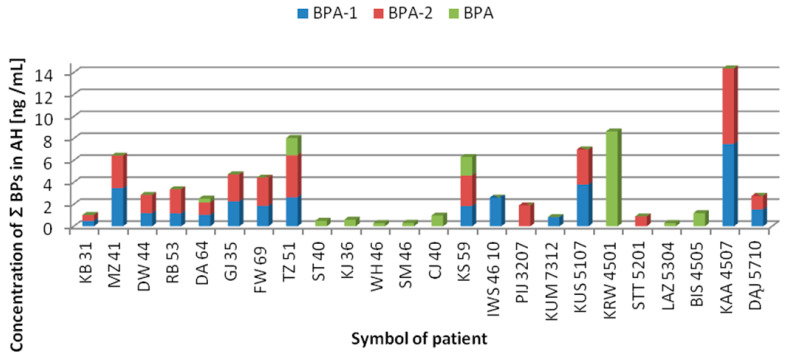
Stacked bar charts dividing each bar into subcategories representing the concentration of BPA-free, and its structural isomers BPA-1, BPA-2 in AH of patients undergoing cataract surgery.

**Figure 6 jcm-11-06402-f006:**
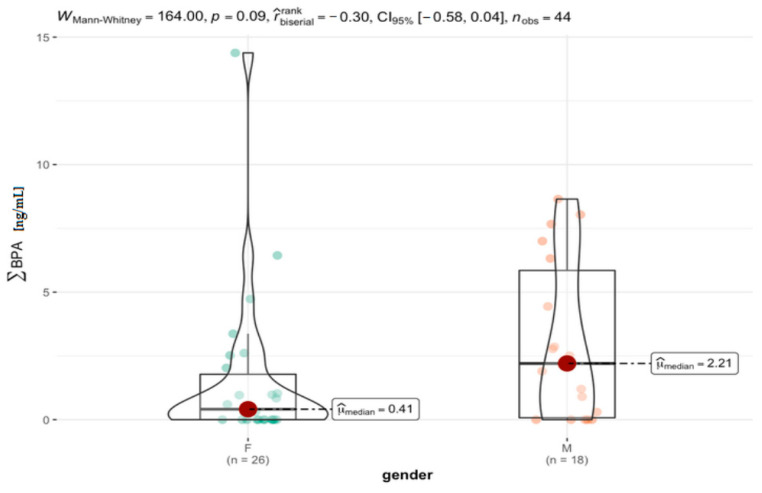
Statistical significance of differences in total BPs content in AH samples of females (F) and males (M).

**Figure 7 jcm-11-06402-f007:**
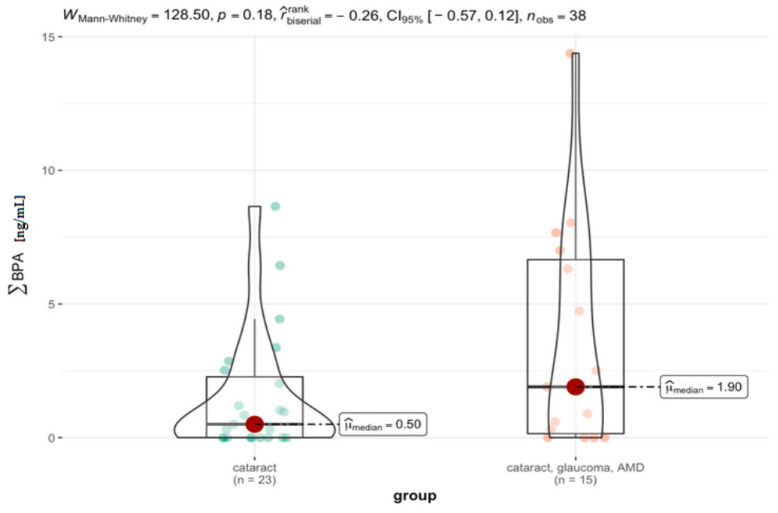
Statistical significance of differences in BPs’ content in AH for cataract and cataract with coexisting ocular pathologies (AMD, glaucoma).

**Figure 8 jcm-11-06402-f008:**
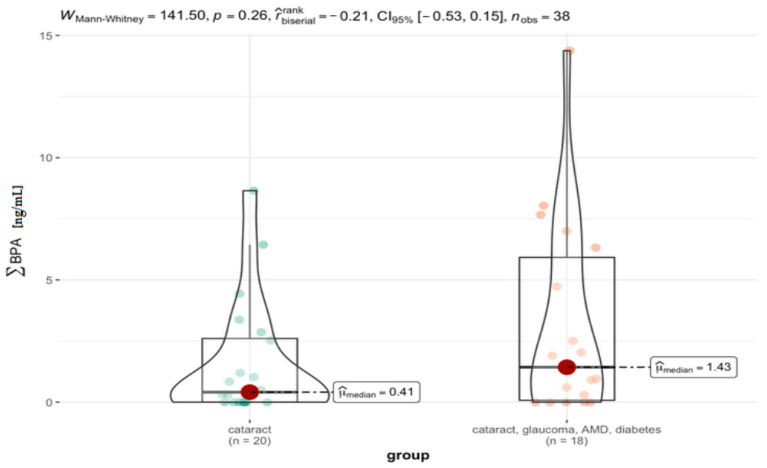
Statistical significance of differences in BPs’ content in AH for cataract and cataract with coexisting ocular pathologies (AMD, glaucoma) and diabetes.

**Table 1 jcm-11-06402-t001:** Demographic characteristic of the patients’ group enrolled in the study.

Group	Gender	*n*	%	Min–Max Age	Median Age	Mean Age ± SD
cases (*n* = 44)	Female	26	59.1	49–92	84.88	77.77 ± 7.12
	Male	18	40.9	53–90	75.00	73.72 ± 8.22

**Table 2 jcm-11-06402-t002:** UHPLC–MS/MS parameters for the multiple reaction monitoring (MRM) acquisition mode.

Compound	Precursor Ion (m/z)	Product Ion (m/z)	DP (V)	EP (V)	CE (V)	CXP (V)
Bisphenol A	227.2	**212.0** 133.0	−130 −130	−10 −10	−24 −32	−16 −16
Bisphenol F	199.2	**93.0** 105.0	−130 −130	−10 −10	−28 −28	−12 −12
Bisphenol A–D_16_ (IS)	241.2	142.1	−130	−10	−40	−16

IS—internal standard; DP—declustering potential; EP—exit potential; CE—collision energy; CXP—collision cell exit potential. The quantifier ion is in bold.

**Table 3 jcm-11-06402-t003:** Parameters obtained for the calibration curves for BPA and BPF.

Compounds	LOD (ng mL^−1^)	LOQ (ng mL^−1^)	Matrix Effect (%)	Working Range (ng mL^−1^)	Determination Coefficient (R^2^)	Calibration Curve
Bisphenol A	0.13	0.248	4.0 ± 1.6%	0.25–20.0	0.987	y = 0.216x + 0.05
Bisphenol F	0.14	0.252	3.9 ± 2.5%	0.25–20.0	0.989	y = 0.291x + 0.01

LOD—limit of detection; LOQ—limit of quantification.

**Table 4 jcm-11-06402-t004:** Intra-assay (*n* = 6) and inter-assay (*n* = 18) precision, and % recovery of BPA I BPF.

Concentration (ng mL^−1^)	Repeatability (RSD_r_, %) (*n* = 6)	Within-Lab Reproducibility (RSD_wR_, %) (*n* = 18)	Expanded Uncertainty (ng mL^−1^)	Apparent Recovery (%)
Bisphenol A				
0.25	3.8 ± 4.1	4.8 ± 4.2	-	103.3 ±3.2
1.0	3.5 ± 3.3	4.5 ± 3.9	1.0 ± 0.17	104.3 ± 2.7
2.5	2.9 ± 2.8	3.6 ± 3.4	-	102.1 ± 2.5
10.0	3.0 ± 3.1	4.0 ± 3.3	-	104.3 ± 4.0
20.0	2.9 ± 3.1	4.1 ± 3.3	-	103.8 ± 3.7
Bisphenol F				
0.25	3.6 ± 4.5	4.6 ± 4.3	-	99.9 ± 3.6
1.0	3.6 ± 3.7	4.1 ± 3.9	1.0 ± 0.22	103.2 ± 3.4
2.5	3.1 ± 3.8	3.9 ± 3.6	-	104.2 ± 3.5
10.0	3.4 ± 3.3	4.1 ± 3.6	-	103.9 ± 3.4
20.0	3.5 ± 3.1	4.2 ± 3.4	-	103.6 ± 3.5

RSD_r_—relative standard deviation for repeatability; RSD_wR_—relative standard deviation for within-lab reproducibility.

**Table 5 jcm-11-06402-t005:** Descriptive statistics for UHPLC–MS/MS measurements of AH samples taken from the entire population of the studied subjects (*n* = 44).

Parameter	BPA-1	BPA-2	BPA-Free
Mean [ng mL^−1^]	0.8809	0.9191	0.3857
Standard error	0.2285	0.2261	0.2021
Median	0	0	0
Mode	0	0	0
Standard deviation	1.5157	1.4998	1.34055
Sample variance	2.2973	2.2494	1.7971
Kurtosis	7.7144	4.6902	35.4541
Skewness	2.4863	2.02203	5.7367
Range	7.51	6.87	8.65
Minimum	0	0	0
Maximum	7.51	6.87	8.65
Sum	38.76	40.44	16.97
Count	44	44	44
DF (*n*)	18	18	12
DF (%)	40.91	40.91	27.27
P25	0	0	0
P50	0	0	0
P75	1.185	1.458	0.3
P90	2.638	2.906	0.866
P99	5.915	5.597	5.657

DF—detection frequency; P25—percentile 25%; P50—percentile 50%; P75—percentile 75%; P90—percentile 90%; P99—percentile 99%.

**Table 6 jcm-11-06402-t006:** Descriptive statistics for female and male groups.

Gender	Vars	*n*	Mean	Sd	Median	Trimmed	Mad	Min	Max	Range	Skew	Kurtosis	Se
female	1	26	1.60	3.08	0.41	0.94	0.61	0	14.38	14.38	2.88	8.72	0.60
male	1	18	3.03	3.16	2.20	2.87	3.27	0	8.65	8.65	0.58	−1.34	0.74

**Table 7 jcm-11-06402-t007:** Descriptive statistics for cataract group and cataract with coexisting ocular pathologies (glaucoma, AMD) group.

Group	Vars	*n*	Mean	Sd	Median	Trimmed	Mad	Min	Max	Range	Skew	Kurtosis	Se
cataract	1	23	1.54	2.30	0.50	1.07	0.74	0	8.65	8.65	1.70	2.12	0.48
cataract, AMD, glaucoma	1	15	3.62	4.27	1.90	3.07	2.82	0	14.38	14.38	1.00	0.03	1.10

**Table 8 jcm-11-06402-t008:** Descriptive statistics for cataract group and cataract with coexisting ocular pathologies (glaucoma, AMD) and diabetes.

Group	Vars	*n*	Mean	Sd	Median	Trimmed	Mad	Min	Max	Range	Skew	Kurtosis	Se
cataract	1	20	1.62	2.44	0.41	1.09	0.61	0	8.65	8.65	1.55	1.42	0.54
cataract, glaucoma, AMD, diabetes	1	18	3.19	4.02	1.43	2.68	2.12	0	14.38	14.38	1.25	0.75	0.95

## Data Availability

Not applicable.

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
