# Peer review of "The First Evidence on the Occurrence of Bisphenol Analogues in the Aqueous Humor of Patients Undergoing Cataract Surgery"

_jcm, 2022, doi:10.3390/jcm11216402_

Round 1
Reviewer 1 Report
This paper is a study on the measurement of BPA in eye fluids, and the number of participants is small, and there are no hypotheses or discussions about exposure sources. It is difficult to determine whether the level of exposure is high or low, and it is difficult to conclude that clinical significance has been found. Unfortunately, this paper is not considered to meet the criteria of publication in this journal.
Author Response
This paper is a study on the measurement of BPA in eye fluids, and the number of participants is small, and there are no hypotheses or discussions about exposure sources. It is difficult to determine whether the level of exposure is high or low, and it is difficult to conclude that clinical significance has been found. Unfortunately, this paper is not considered to meet the criteria of publication in this journal.
The authors thank the reviewer for reading our article and for the time devoted to this work. We agree with the reviewer that this is not a population study. When we started the research, we were not 100% sure that bisphenols would be found in this biological material at all, therefore our work should be treated as a preliminary study. We would like to point out that we could not apply to the bioethical commission for permission to collect more samples in preliminary tests, because it would be unethical. Now, knowing that bisphenols are getting into the eye fluid, these tests can be continued. We hope that other researchers, especially ophthalmologists, will also be interested in this topic.
Regarding the sources of exposure, it is known that exposure to bisphenols is widespread and unavoidable. Regarding the routes of exposure, studies are primarily conducted on animals and not on humans. As for the degree of exposure, it is very difficult to verify in the case of humans. Cataract surgery is performed on elderly people who are often unaware of environmental threats. However, we agree with the reviewer that the issue is interesting and should be analyzed on the basis of a special survey. Such environmental studies, however, were not the subject of this work.
Reviewer 2 Report
1. Abstract : Authors do not analyze the effect of bisphenols on visual function but just the presence of this substances in aqueous humor – please amend
2. Abstract: please use term: aqueous humor instead of fluid located in the anterior chamber – it sounds awkward, especially for ophthalmologists
3. Lines 46-49 – please provide references for described previous studies
4. 3.1 and 3.2 paragraphs – I am not a chemist, however my perception is that these paragraphs would better suit in the methods section. Please amend or provide explanation.
5. Lines 278-285 – these important data should be presented in the discussion section as the comment to author’s results
6. Paragraph : “statistical analysis” should be renamed according to its contents (comparison between the specific groups)
7. There are mistakes in table’s numbering. Please amend.
8. Table 3 on page 10 (should be 6): glaucoma not glaucoma
9. Lines 323-325 – relationship between HBP and ocular disorders is not that straightforward. Please soften the tone. Besides, I would not use the expression: oculopathy
10. Line 352 : hypertensive angiopathy or retinopathy
11. Discussion- general remark: authors should concentrate more on interpretation of their results. Presented data are interesting, however they go beyond the subject of the paper. Please shorten the discussion section.
12. Conclusions: please be more brief. Do not repeat the results , just formulate the most important conclusions.
Author Response
The authors would like to thank the reviewer for valuable comments that allowed us to improve our work. We greatly appreciate the effort and time devoted to us.
Abstract : Authors do not analyze the effect of bisphenols on visual function but just the presence of this substances in aqueous humor – please amend.
Thank you for this suggestion. It was corrected.
- Abstract: please use term: aqueous humor instead of fluid located in the anterior chamber – it sounds awkward, especially for ophthalmologists.
Yes. It was done.
- Lines 46-49 – please provide references for described previous studies
Appropriate new references were added [4-6].
- 3.1 and 3.2 paragraphs – I am not a chemist, however my perception is that these paragraphs would better suit in the methods section. Please amend or provide explanation.
Yes. It was our mistake. Section: Materials and Methods belong to the second, not the third paragraph. It was corrected.
- Lines 278-285 – these important data should be presented in the discussion section as the comment to author’s results.
Yes. Thank You for this suggestion. Indeed, this part belongs to the discussion. We transferred highlighted lines from Results into the Discussion chapter.
- Paragraph : “statistical analysis” should be renamed according to its contents (comparison between the specific groups).
Yes. Thank You very much for this suggestion. We corrected the subchapter title.
- There are mistakes in table’s numbering. Please amend.
We are very sorry for these mistakes. We corrected all inconsistencies in numbering.
- Table 3 on page 10 (should be 6): glaucoma not glaucoma
We corrected the above mistakes.
- Lines 323-325 – relationship between HBP and ocular disorders is not that straightforward. Please soften the tone. Besides, I would not use the expression: oculopathy.
Thank You for this suggestion. Absolutely, we agree with the reviewer. The sentence has been corrected.
- Line 352 : hypertensive angiopathy or retinopathy
Yes. Thank You for this suggestion. It was corrected.
- Discussion- general remark: authors should concentrate more on interpretation of their results. Presented data are interesting, however they go beyond the subject of the paper. Please shorten the discussion section.
The discussion has been shortened. Some part has been transferred to the introduction to give the whole background of the research, our initial intentions, and inspirations. We do hope that it will be interesting for researchers focused on this subject. However, in the discussion, we left some information about the possible impact of BPA on the functioning of the organ of vision. We distinguished those pathways that are most prone to dysregulation under the influence of BPA. The presented threats justify the need for further research in this direction. However, if the reviewer believes it is necessary, we can shorten part of the discussion even further.
- Conclusions: please be more brief. Do not repeat the results , just formulate the most important conclusions.
Yes. Thank You. We improved our conclusion. We do hope it is better now.
Reviewer 3 Report
The study seems interesting and well written, however, the abstract does not point out the discussion and the conclusion sections.
In addition, lanes 302-306 should be moved to the methods section, with a paragraph for statistical analysis
Author Response
The study seems interesting and well written, however, the abstract does not point out the discussion and the conclusion sections.
In addition, lanes 302-306 should be moved to the methods section, with a paragraph for statistical analysis
Thank You very much for this suggestion. We transferred highlighted sentence to the Methods chapter. The discussion section has been shortened according to other suggestions. The final conclusion has been added to the Abstract part.
Reviewer 4 Report
Recently a plastic pollution of the environment has become a great problem worldwide. It leads to a widespread exposure of individuals to bisphenols (BPs)– plastic additives that affect a variety of physiological functions as it was demonstrated on many animal models.
I have read a paper by Flieger et al., with a great interest. This paper is the first to describe the presence of bisphenols (BPA and BPF) in aqueous humour (AH) of patients underwent the cataract surgery. The authors analysed the AH of 26 women and 18 men with cataract, and in 22 of them comorbidities were diagnosed; DM in 7, AMD in 10 and Glaucoma in 5. The authors found in 40.9% of samples detectable levels of BPA isomers: BPA-1 and BPA-2. They demonstrated the different concentration of BPAs in AH among patients with cataract and with cataract and coexisting diseases.
However, as it is known that exposure to bisphenols is inevitable due to contaminated food, and via transdermal and inhalation routes, it would be interesting to complete the study with the analysis of the factors that potentially may be responsible and may influence the AH levels of these substances, such as: dietary habits, occupation, living environment.
Author Response
Recently a plastic pollution of the environment has become a great problem worldwide. It leads to a widespread exposure of individuals to bisphenols (BPs)– plastic additives that affect a variety of physiological functions as it was demonstrated on many animal models.
I have read a paper by Flieger et al., with a great interest. This paper is the first to describe the presence of bisphenols (BPA and BPF) in aqueous humour (AH) of patients underwent the cataract surgery. The authors analysed the AH of 26 women and 18 men with cataract, and in 22 of them comorbidities were diagnosed; DM in 7, AMD in 10 and Glaucoma in 5. The authors found in 40.9% of samples detectable levels of BPA isomers: BPA-1 and BPA-2. They demonstrated the different concentration of BPAs in AH among patients with cataract and with cataract and coexisting diseases.
However, as it is known that exposure to bisphenols is inevitable due to contaminated food, and via transdermal and inhalation routes, it would be interesting to complete the study with the analysis of the factors that potentially may be responsible and may influence the AH levels of these substances, such as: dietary habits, occupation, living environment.
The authors thank you very much for this assessment, and especially for pointing out that our work proves the presence of bisphenols in the eye fluid for the first time. We hope that popularizing our work will draw the attention of the ophthalmologists community to this new threat to the health of the eye organ.
The authors would like to thank the reviewer for the suggestion of further research devoted to the analysis of the factors responsible for the contamination of body fluids with bisphenols. Such studies would require close collaboration with healthcare professionals. An appropriate questionnaire should be created and the patient population expanded. Certainly, an in-depth analysis of the questionnaires would reveal which eating habits or the living environment contribute to the poisoning of the human body with bisphenols. We will continue our work and we hope that other researchers interested in the topic will also join us.